# Evaluation of Vortex Generators in the Heat Transfer Improvement of Airflow through an In-Line Heated Tube Arrangement

**Syaiful \*, Tri Wahyuni, Bambang Yunianto and Nazaruddin Sinaga**

Department of Mechanical Engineering, Engineering Faculty, Diponegoro University, Semarang 50275, Indonesia; wahyunitri2910@gmail.com (T.W.); b_yunianto@undip.ac.id (B.Y.); nazarsinaga@yahoo.com (N.S.)
\* Correspondence: syaiful@lecturer.undip.ac.id; Tel.: +62-8122-8501-462

**Abstract:** Improving heat transfer from surface to airflow is a current research concern for enhancing energy efficiency. The use of vortex generators for improving heat transfer from the surface to the airflow is very effective. Therefore, this study focuses on applying flat and concave vortex generators with and without holes in order to improve heat transfer. In this study, the number of pairs of vortex generators was varied from one to three pairs at a certain angle of attack for various forms of vortex generators. The airflow velocity through the duct was varied in the range of 0.4 to 2.0 m/s at 0.2 m/s intervals. From the investigation results, we observed that the highest thermal performance was found with the use of concave delta winglets without holes for various pairs of vortex generators in terms of the overall Reynolds number. The highest thermal enhancement factor was found to be around 1.42 at a Reynolds number of approximately 9000. From this study, it was also shown that the lowest cost–benefit ratio was about 1.75 at a Reynolds number of approximately 3500 for three pairs of vortex generators.

**Keywords:** vortex generator; heat transfer improvement; longitudinal vortex; thermal performance; pressure drop





## 1. Introduction

Fin and tube condensers are widely used in household air conditioning, power generation, and the chemical industry. Therefore, the improvement of the heat transfer on the air side of the condenser is necessary in order to enhance energy efficiency. The low thermal conductivity of the air side is the main reason for improving the heat transfer of fin and tube condensers. The application of vortex generators (VGs) is a passive method for increasing the heat transfer rate. Furthermore, the use of VGs for enhancing heat transfer in order to improve system efficiency to reduce costs has attracted the attention of many researchers.

Various researchers have carried out studies on the improvement of heat transfer using passive methods [1–5]. Li et al. [6,7] numerically and experimentally studied the effect of using a vortex generator on increasing heat transfer in a finless heat exchanger. They found that using a double-triangle LVG increases the heat transfer coefficient more than that of a common LVG. LVG is produced by the VG winglet, which can reduce the thickness of the boundary layer, which impacts the heat transfer [8]. The longitudinal vortex (LV) produced by the winglet-type VG can enhance heat transfer better than the transverse vortex (TV) produced by the wing-type VG [9]. This is due to the three-dimensional flow from the LV with the rotation axis in the direction of the main flow, resulting in a better mixing of the flow than that from the TV [10]. Mixing the cold fluid flow away from the wall with the hot fluid flow near the wall results in increased heat transfer [11].

The heat transfer improvement using VG was better than that gained using twisted tapes, helical wire coils, and conical rings because of the lower pressure loss of the VG, as observed by Wang et al. [12]. Their work demonstrates that VG was able to enhance heat

transfer with a low pressure drop. A comparative study on the process of improving heat transfer by applying various rectangular winglet (RW) VGs to fin and tube heat exchangers was carried out by Ashish et al. [13]. They found that up-wavy RWVGs showed better heat transfer improvements than those of other RWVGs. The use of curved rectangular VGs shows an effective heat transfer improvement, as already reported by Lu and Zai. [14]. They noted that VGs with a 0.25 curvature showed the best heat transfer improvement. This indicates that the geometry and arrangement of the VG mounting has an impact on the thermal–hydraulic performance. Pongjet and Sompol investigated the thermal–hydraulic performance of a heat exchanger that mounts a V-shaped VG winglet [15]. They found that V-shaped rectangular winglet vortex generators (RWVGs) exhibit increased heat transfer rates and higher pressure drops compared to those of V-shaped delta winglet vortex generators (DWVGs). Various geometric shapes of VGs were investigated by Li et al. (2020) with regard to their ability to improve heat transfer in a helical channel [16]. The authors refer to these as a streamlined winglet pair (SWP). They combined an SWP with a rectangular cross-section to achieve increased heat transfer and obtained an increase of up to 46% compared to a formation with no VG. The advantages and increased efficiency of triangular-winglet VGs for improving heat transfer were studied by Tian et al. [17]. They investigated the application of triangular-winglet VGs for enhancing heat transfer rates in various channel geometries. Various studies related to the improvement of heat transfer have focused on several study parameters, such as the effect of VG geometry, angle of attack, aspect ratio, and arrangement of VGs on flow, etc. (see Refs. [18–29]).

The high pressure drop resulting from the use of a curved-winglet VG encouraged Syaiful et al. (2019) to investigate the application of perforated concave delta winglet (PCDW) VGs for enhancing thermal–hydraulic performance [30]. They found a decrease in pressure drop with lower heat transfer rates when using PCDW VGs compared to when using CDW VGs. Although the results of their study were not as expected, they showed a significant decrease in pressure drop. In a subsequent study, Syaiful et al. (2021) tried to investigate the use of concave rectangular winglet (CRW) VGs for increasing the heat transfer rate, where CRW VGs showed a better thermal improvement than concave delta winglet (CDW) VGs [31]. As with previous studies, they found that pressure drops could be reduced with the use of perforated VGs compared with non-perforated VGs.

From the various studies that have been carried out, it can be stated that evaluating the profits of using perforated concave delta winglets (PCDW) VGs for improving heat transfer from the surface to one row of tubes arranged in-line remains a topic that has rarely been investigated. Therefore, the novelty of the present work is the use of PCDW VGs to improve the rate of heat transfer from the tube surfaces to the air flow in a rectangular channel. The thermal–hydraulic performance and advantages of PCDW VGs were evaluated by determining the variables of temperature and pressure drop of the flow from the experimental test. The thermal–hydraulic performance and profit of PCDW VGs are indicated by the thermal enhancement factor (TEF) and cost benefit ratio (CBR). Furthermore, flow visualization was performed to determine the longitudinal vortex (LV) generated by VGs. The evaluation of the uncertainty in the study was also observed by determining the error of the calculated parameters.

In this work, validation was carried out by comparing the results of the Whitaker study and the current study, as discussed in Ref. [32]. The temperature data obtained at a predetermined velocity were used to calculate the Nusselt number. Meanwhile, the pressure drop data obtained using a micromanometer at various flow rates were used to determine the coefficient of friction. The Nusselt number and friction coefficient data were used to determine the thermo-hydraulic performance (TEF).

## 2. Materials and Methods

### 2.1. Experimental Set-Up and Procedures

Figure 1 illustrates the experimental set-up of the current work. The use of PCDW and CDW VGs to increase the rate of heat transfer from in-line tubes to the airflow in the

rectangular channel was investigated in this experiment. The channel had a length, height, and width of 370 cm, 18 cm, and 8 cm, respectively. This rectangular channel was made of glass with a thickness of 1 cm. Air was sucked in by a blower (Wipro with a rated voltage of 380 V–50 Hz) located at the end of the channel. The inlet airflow velocity was regulated using an inverter (Mitsubishi Electric-type FR-D700 with an accuracy of ±0.01). Air flowed through a straightener, which consisted of an array of aluminum pipes with a diameter of 5 mm and a length of 290 mm. The wire mesh was placed directly behind the straightener to achieve a uniform flow. Airflow velocity was measured with a hotwire anemometer (Type AM-4204 made by Lutron with an accuracy of ±0.1) placed 27 cm from the wire mesh. In this experiment, the airflow velocity was varied in the range of 0.4 m/s to 2.0 m/s in intervals of 0.2 m/s. Then, air flowed through the test specimen, which consisted of six tubes in an in-line row heated at a constant rate of 40 W. The six tubes of the test specimens were mounted on a plate.

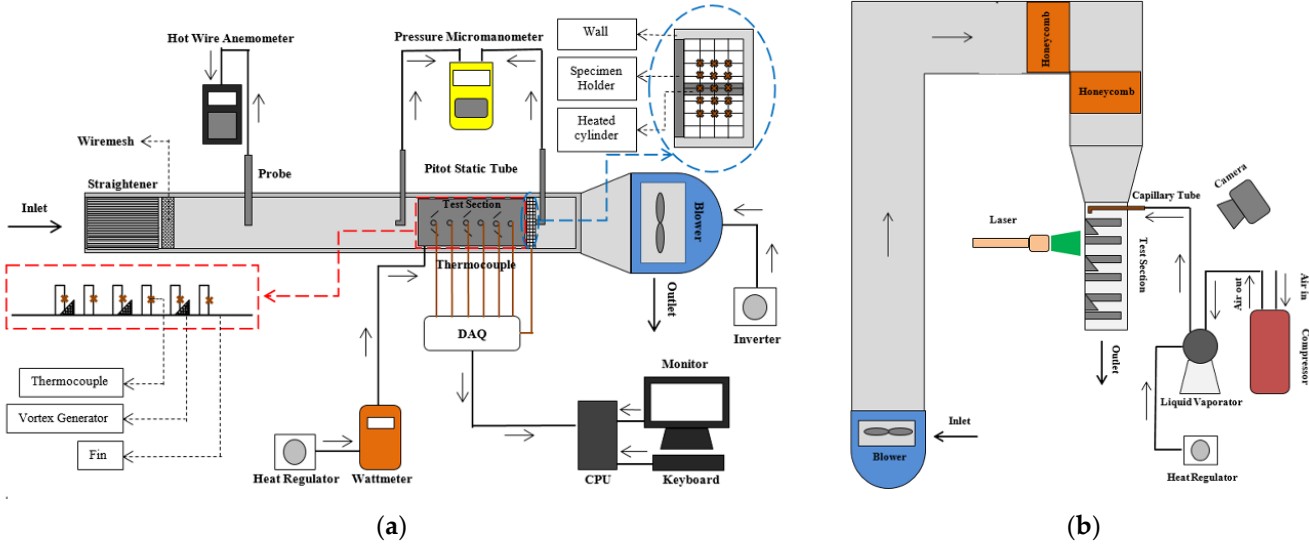

**Figure 1.** (**a**) Schematic of heat transfer rate and pressure drop test equipment; (**b**) flow visualization experimental set-up.

In order to obtain the thermal–hydraulic performance in this experiment (see Figure 1a), six tubes were heated at a rate of 40 W, which was regulated by a heating regulator and monitored by a Wattmeter (Lutron DW-6060 with an accuracy of ±0.01). Several thermocouples (type K with a temperature range of −200–1250 °C and accuracy of ±0.001) were installed at the inlet, on the surface of each tube, and on the plate where the tubes were attached, with 15 thermocouples placed on the outlet side. Then, the thermocouples were connected to the data acquisition device (Advantech USB-4718 type with an accuracy ±0.001), which was connected to the computer CPU for recording and monitoring. In order to obtain pressure-drop data, two pitot tubes were placed 26 cm in front of the test section and 2.5 cm behind the test section. The two pitot tubes were connected to a micromanometer (Fluke type 922 with an accuracy of ±0.05) to monitor the pressure drop of the airflow through the test section. The pressure drop was recorded every five seconds for each velocity.

In the present study, flow visualization was performed to observe the LV produced by VGs. The experimental set-up for flow visualization is shown in Figure 1b. In this flow visualization experiment, the compressor was used to pump the steam produced by a liquid vaporizer through a capillary tube inside the channel where the test section was installed. This smoke would flow along with the main flow through the test section inside the channel. LV was observed as the flow passed through the VGs and was captured by the rectangular plane formed by the laser firing on the solid glass cylinder. A digital camera was used to record and monitor the flow structure and LV produced by the VGs.

*2.2. Test Specimen*

Figure 2 provides the detailed geometry of the test specimens and the PDW and PCDW VG. The test specimen consisted of six tubes with a diameter of 19 mm and height of 60 mm; the tubes were mounted on a plate with a length, width, and thickness of 500 mm, 165 mm, and 1 mm, respectively. In this study, the DWP, CDWP, PDWP, and PCDWP VGs were evaluated in a common flow down (CDW) configuration. The distance between the leading edge of the VG and the center of the cylinder was 1.5 mm. The six tubes were arranged in-line with a distance between the centers of the tubes of 60 mm. The attack angle of the VGs was determined to be 20° from the main flow direction. Tests were carried out for one, two, and three pairs of VGs arranged in-line. The leading-edge distance between a pair of VGs and another pair was 120 mm. The perforated VGs had a length of 30 mm and a height of 30 mm, with 15 holes with diameters of 2.5 mm. The distance between the holes was 5 mm. Figure 3 shows a photo of the test specimen in this study for the three pairs of PDW and PCDW. In total, 13 specimens were tested in this work, consisting of three variations in the number of pairs of DW, CDW, PDW, and PCDW VGs as well as a baseline.

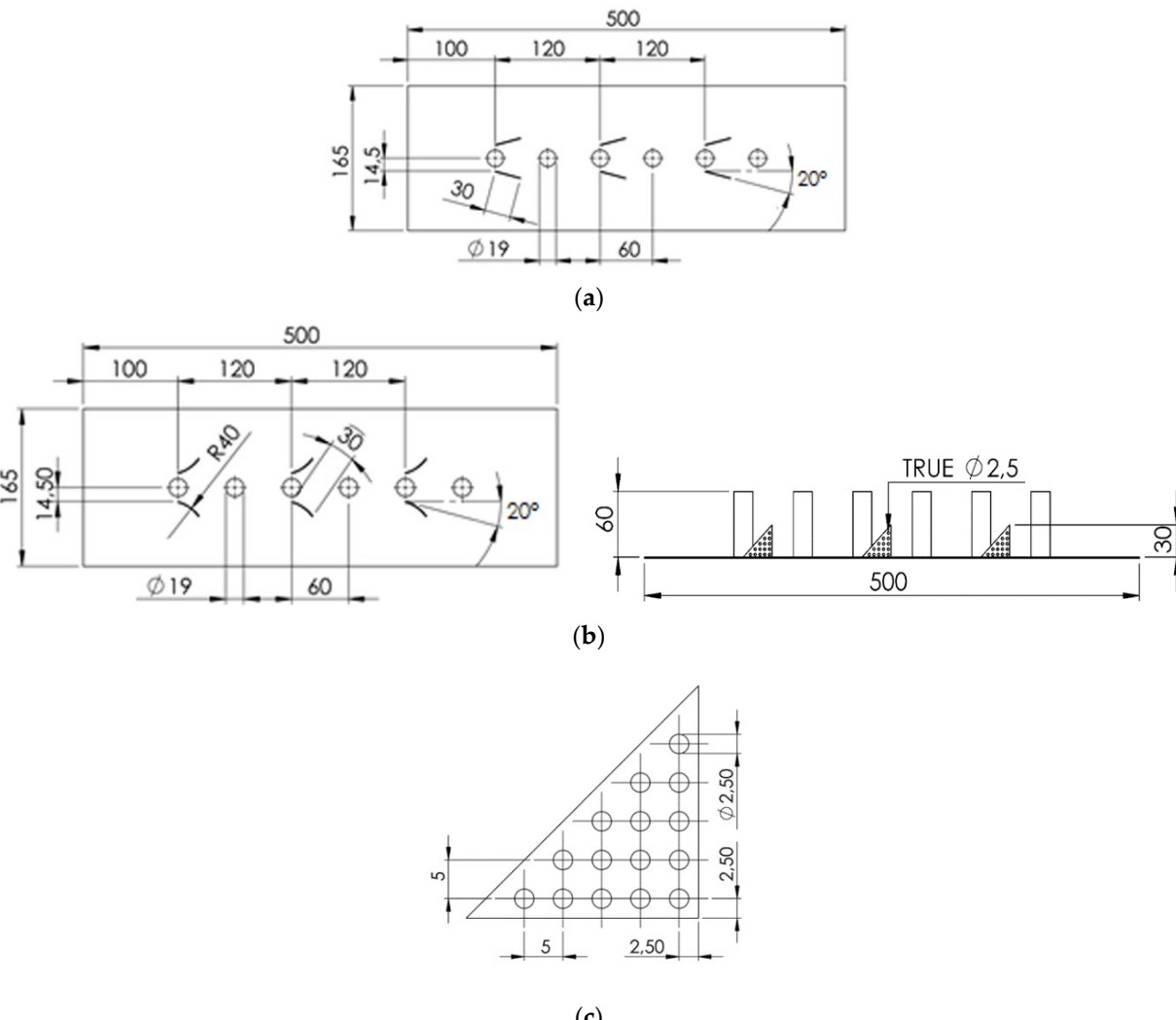

**Figure 2.** Geometry of the test specimen: (**a**) top view of three pairs of in-line PDW and PCDW VGs; (**b**) side view of three pairs of in-line PDW and PCDW VGs; (**c**) geometry of the perforated VG.

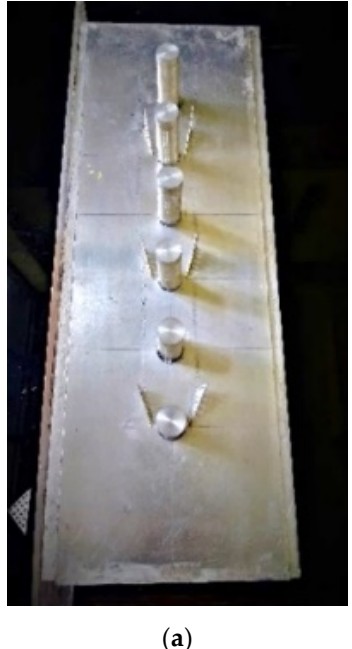 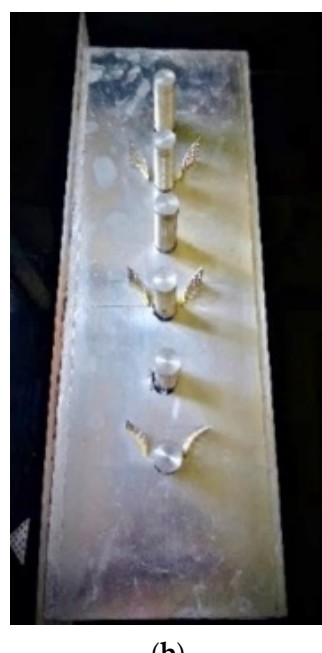

| (**a**) | (**b**) |

**Figure 3.** Test specimen: (**a**) three pairs of PDW VGs and (**b**) three pairs of PCDW VGs.

*2.3. Parameter Definition*

Several equations were used in this study to evaluate thermo-hydraulic performance. The convection heat transfer coefficient was determined using Equation (1) [16,18]:

$$h = \frac{N_u \, k}{D_h} \tag{1}$$

where $N_u$, $k$, and $D_h$ are the Nusselt number, fluid thermal conductivity, and hydraulic diameter $\left(D_h = \frac{2(W \times H)}{(W+H)}\right)$, respectively. Nusselt number can be calculated using Equation (2) [16,18].

$$N_u = \frac{Q}{A \, LMTD} \cdot \frac{D_h}{k} \tag{2}$$

where $Q$ is the heat transfer from the tube surface to the fluid flow, $A$ is the tube surface area, and *LMTD* is the logarithmic mean temperature difference of the fluid. The values of $Q$ and *LMTD* can be calculated using Equations (3) and (4), respectively [16,18].

$$Q = \dot{m} \, C_p \, (T_{out} - T_{in}) \tag{3}$$

$$LMTD = \frac{(T_w - T_{out}) - (T_w - T_{in})}{ln((T_w - T_{out})/(T_w - T_{in}))} \tag{4}$$

where $\dot{m}$, $C_p$, $T_{out}$, and $T_{in}$ are the fluid mass rate, fluid specific heat, outlet, and inlet temperature of the fluid, respectively.

The hydrodynamic performance was calculated by measuring the pressure drop of the flow at the inlet and outlet sides of the test specimen. Then, the friction factor was calculated using Equation (5) [33].

$$f = \frac{2 \, \Delta P \, D_h}{\rho \, V2 \, (L + 6D)} \tag{5}$$

where $\rho$, $V$, and $L$ are the density of the air, the air flow velocity, and the length of the test specimen plate, respectively.

From the thermal and hydraulic performance determined by Equations (2) and (5), respectively, the thermal–hydraulic performance can be calculated using Equation (6) [33]:

$$\text{TEF} = \frac{Nu/Nu_0}{\left(\frac{f}{f_0}\right)^{1/3}} \tag{6}$$

Another important parameter used in the heat exchanger analysis is the cost per benefit ratio (CBR), which is the ratio of the percentage of pressure drop to the percentage of Nusselt number between set-ups using VGs and those not using VGs [34]. This CBR indicates whether the method used to increase the heat transfer rate is economical or not. If the CBR value is below one, then the method used to improve the heat transfer is efficient. CBR can be calculated using Equation (7) [34].

$$\text{CBR} = \frac{\%\Delta P}{\%Nu} \tag{7}$$

where %$\Delta P$ and %$Nu$ are the percentage difference of the pressure drop and Nusselt number for the case using VG with that of without using VG.

### 2.4. Validation

Validation was necessary to ensure that the experiments carried out were going in the right direction. This validation was carried out by comparing the current experiment with the experiment from Whitaker, for which validation results can be seen in previous studies [31]. In the validation, the current experimental conditions were adjusted to the experimental conditions used by Whitaker. The value of the Nusselt number was determined to be in the Reynolds number range from 2000 to 10,000. The overestimated values found from the present experimental results were compared with those from Whitaker's experiments, as the heat induced was higher than that in Whitaker's work. However, the Nusselt number of the experimental results now shows a similar tendency according to Whitaker's experimental results.

## 3. Results and Discussion

### 3.1. Effect of VG on Heat Transfer

The heat transfer improvement due to the installation of VGs can be represented by the value of the Nusselt number ratio of the set-ups with and without VGs, as shown in Figure 4. Figure 4 shows the ratios of the Nusselt numbers for mounting DWP, CDWP, PDWP, and PCDWP VGs for one, two, and three pairs of VGs to Reynolds numbers. From this study, it was found that the addition of pairs of VGs was able to increase the rate of heat transfer, as indicated by the increase in the Nusselt number ratio with the rise in the number of pairs of VGs at the same Reynolds number. The main reason for this is that the addition of pairs of VGs increases the number of LVs generated and strengthens the LV [18]. A high strength of LV results in good fluid mixing, which results in increased heat transfer [30].

From Figure 4, it can be observed that the CDWP VGs set-up without holes shows the highest Nusselt number ratio at the same number of pairs of VGs and the same Reynolds number. The highest Nusselt number ratio for the case of VGs without holes found in the installation of one, two, and three pairs of CDW VGs is 1.64 at Re = 8973, 1.65, and 1.73 at Re = 4300. As for the case of perforated VGs, the highest Nusselt number ratio for the use of one, two, and three pairs of CDW VGs was found to be 1.56, 1.57, and 1.62 at Re = 4300. From the experimental results shown in this study, it can be concluded that the hole in the VG has an impact on a slight decrease in the Nusselt number ratio. This is because the jet flow formed from the VG hole is able to damage the LV produced by the VG, meaning that its intensity is reduced, which results in a decrease in the fluid mixing and the Nusselt number ratio afterward [35]. The Nusselt number ratios in cases with one, two, and three pairs of DW and PCDW were found to be almost the same for all Reynolds numbers.

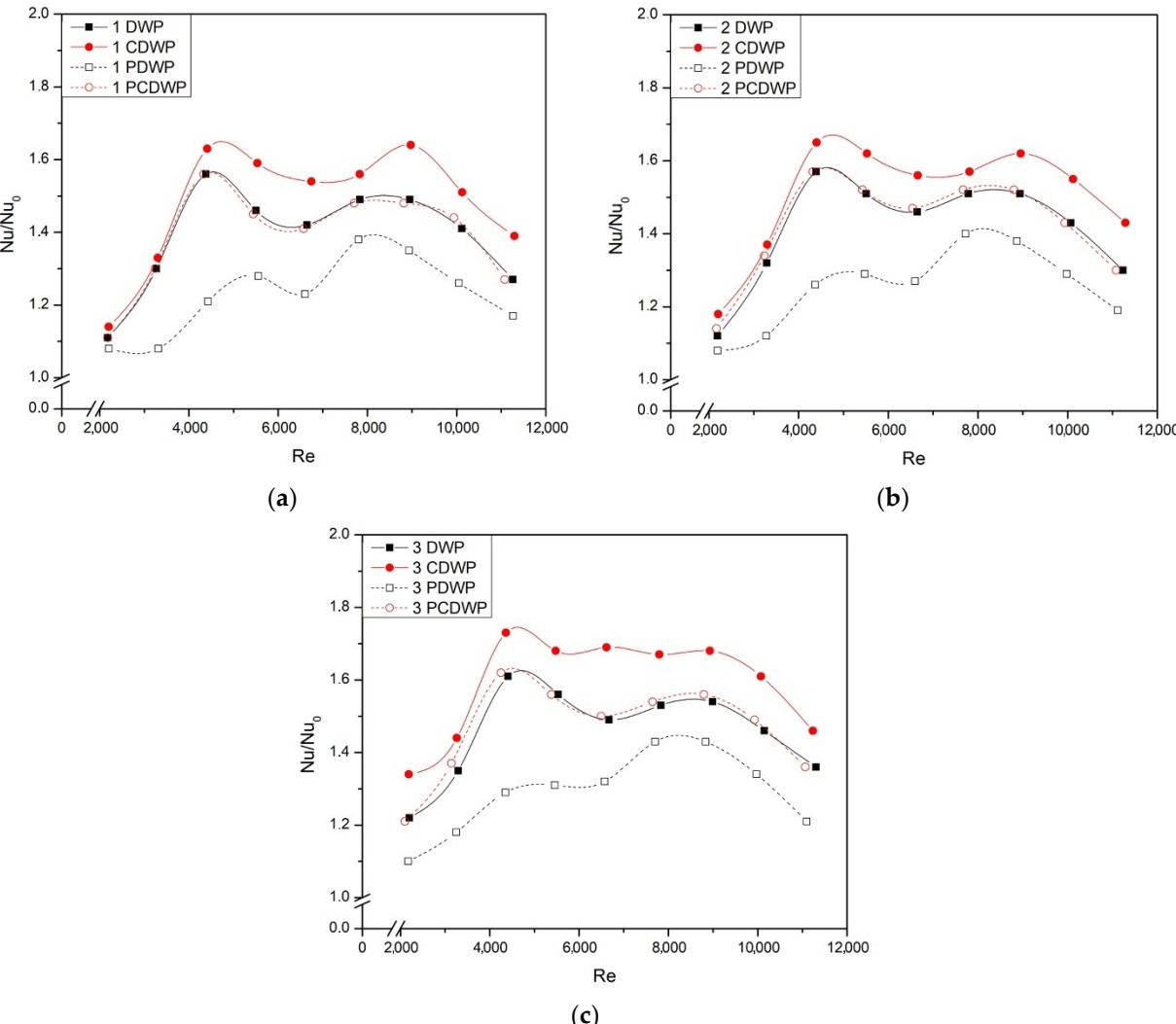

**Figure 4.** Nu/Nu$_0$ on the use of VGs with/without holes on the Reynolds number and the number of pairs of VGs. (**a**) One pair of VG, (**b**) Two pairs of VG, and (**c**) Three pairs of VG.

### 3.2. Effect of VG on Friction Factor

Figure 5 provides a comparison of the friction factor ratios ($f/f_0$) for the installation of DWP, CDWP, PDWP, and PCDWP VGs at various Reynolds numbers. From the current experimental study, in general, it was found that the holes in the VGs were able to reduce the friction factor ratio, which indicates a decrease in pressure drop with the holes in the VGs. This decrease in pressure drop is caused by a decrease in the frontal area through which the flow passes, as well as the weakening of the LV. The jet flow effect of the VG holes reduces the stagnation of flow, reducing the local pressure losses [36]. A higher $f/f_0$ was found in the CDWP installation than in the other cases due to the large frontal area of the CDWP VGs. The highest $f/f_0$ for the holeless VG case was found to be 1.6, 1.8, and 2.8 at Re = 4300 for one, two, and three pairs of CDW VGs, respectively. From Figure 5, it can be observed that the highest $f/f_0$ for all cases in one, two, and three pairs was found at a Reynolds number of around 4400. From Figure 5, it can also be seen that $f/f_0$ in cases one, two, and three of DWP and PCDWP shows almost the same value. $f/f_0$ tends to decrease with respect to the Reynolds number.

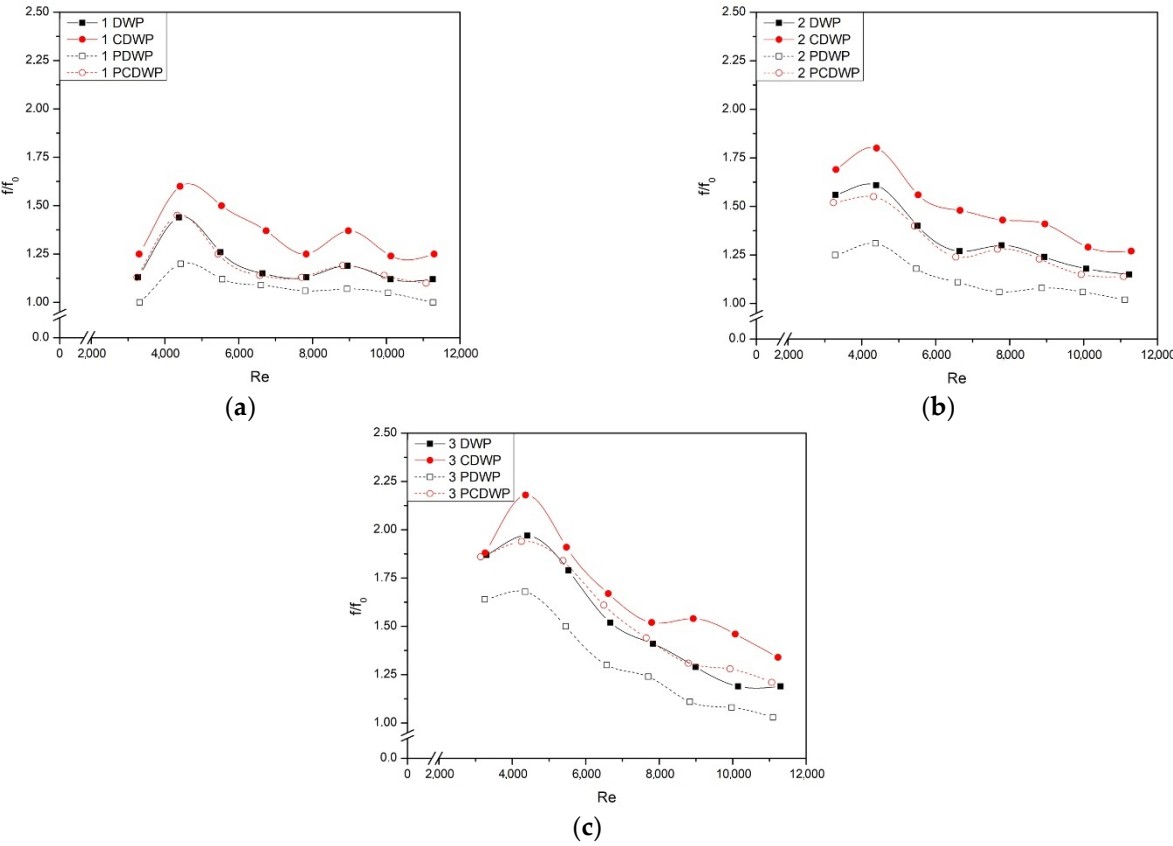

**Figure 5.** $f/f_0$ for the use of VGs with/without holes on the Reynolds number and the number of pairs of VG. (**a**) One pair of VG, (**b**) Two pairs of VG, and (**c**) Three pairs of VG.

### 3.3. Effect of VG on Thermal–Hydraulic Performance

The thermal–hydraulic performance during the use of VGs can be represented by TEF. A TEF of higher than one indicates that there was an effective heat transfer improvement. Figure 6 shows that the highest thermal–hydraulic performance was found with the use of CDWP VGs. This indicates that the heat transfer improvement caused by the use of CDWP VGs was more significant than the increase in pressure drop. The use of CDWP and PCDWP VGs resulted in a large LV radius with a high intensity, which reduced the wake region [37]. The reduction in the wake region increased the flow velocity downstream of the tubes and reduced the recirculation zone, leading to an increase in the rate of heat transfer in that region [18,38]. A decrease in the wake region reduced the stagnation region, which reduced the pressure drop [38].

As shown in Figure 6, the maximum TEF for cases with one, two, and three pairs of PDW VGs is reduced by 5.9%, 4.4%, and 2.2%, respectively, compared to that of DW VGs. Meanwhile, in the case of concave VGs, the maximum TEF is reduced by 5.8%, 2.1%, and 2.8% for cases with one, two, and three pairs of PCDW VGs compared to that of CDW VGs. In general, the highest TEF was observed to be 1.45 with the installation of three pairs of CDW VGs at Re = 8928. The TEF tended to increase at Reynolds numbers below 8300 and decrease thereafter. The TEF showed almost the same values for cases one, two, and three of DWP and PCDWP.

### 3.4. Effect of VG on the Economic Value of Improved Heat Transfer (CBR)

The cost–benefit ratio (CBR) is a parameter used to evaluate the economics of a heat transfer improvement method [34]. The effect of the type of VG used and the number of pairs of VGs on the CBR at various Reynolds numbers is shown in Figure 7. With the installation of one pair of VGs, the lowest CBR was found in the case of PDWP for the

same Reynolds number. This indicates that the installation of PDWP VGs is economically the best out of all the VG installations. However, the CBR, in this case, reached its highest value of around 0.965 at Re = 4400. With the installation of two pairs of VGs, PDWP still occupied the lowest value at Re numbers greater than 5500. Below Re = 5500, PCDWP VGs had the lowest CBR value. Furthermore, the lowest CBR was found with the use of PDWP in the installation of one, two, and three pairs of VGs at Re numbers greater than 6600. The increase in the number of VG pairs resulted in an increase in the CBR at the same Reynolds number for Reynolds numbers below 6000.

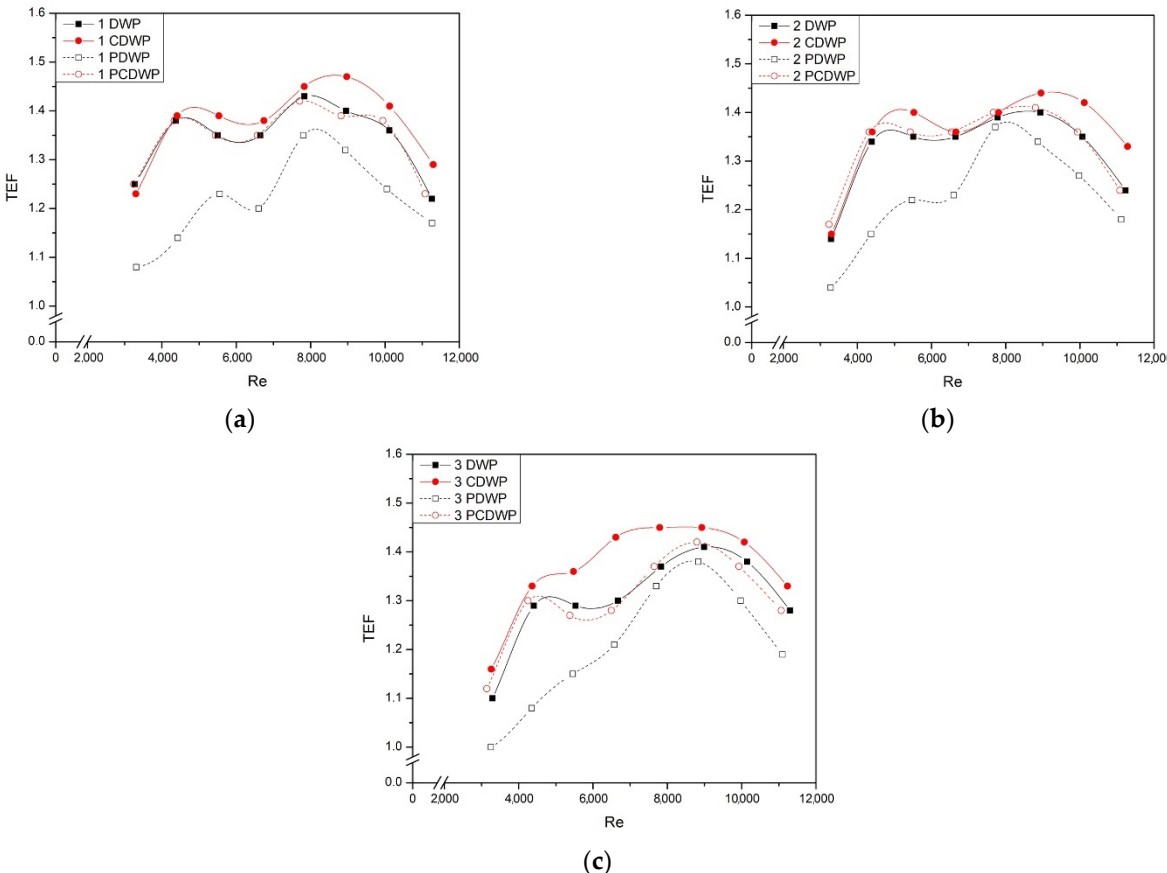

**Figure 6.** TEF for various VGs with/without holes against Reynolds number for the use of various numbers of VG pairs. (**a**) One pair of VG, (**b**) Two pairs of VG, and (**c**) Three pairs of VG.

### 3.5. Flow Visualization

A flow visualization test was carried out to determine the LV structure formed by the installation of VGs. Figure 8 shows the LV formed after the flow passed through the perforated VGs. The LV formed was captured by a cross-section plane of the flow created by a laser beam shooting into a transverse glass cylinder. The position of the cross-section plane was at x/L = 0.28, where x is the distance from the plane to the inlet end of the test plate and L is the total length of the test plate. The flow structure in the downstream region of the perforated VGs in the flow direction is shown in Figure 9. This visualization was performed with the use of one pair of PDW and PCDW VGs.

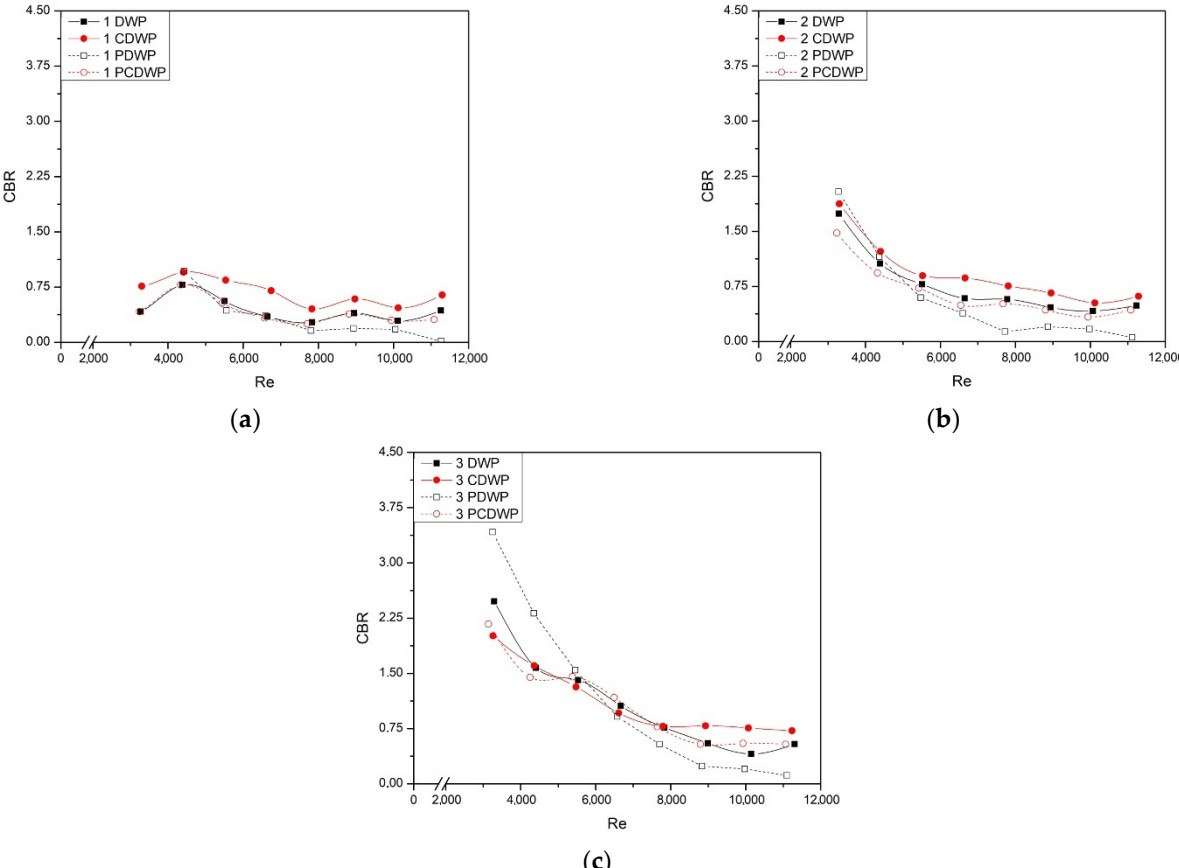

**Figure 7.** CBR showing the effect of the use of VG with/without holes on the variation in the Reynolds numbers and the number of pairs of VGs. (**a**) One pair of VG, (**b**) Two pairs of VG, and (**c**) Three pairs of VG.

From Figure 8, it can be observed that the LV generated by the PCDWP VGs had a larger radius than that of the PDWP VGs. The large LV radius has an impact on the thinning of the thermal boundary layer on the surface, resulting in an increase in the rate of heat transfer from the surface to the flow [34]. In addition, the use of a large LV pushed the wake region, which could enhance the heat transfer in the area [37]. The large size of the LV produced by the PCDWP VGs was caused by the instability of the centrifugal force when the flow passed through the concave surface [30]. This indicates that the heat transfer improvement brought about by installing PCDWP VGs is better than that brought about by PDWP VGs.

From Figure 9, it can be observed that the use of PCDWP VGs generates LV by occupying a wider flow area compared to that of PDWP VGs in the downstream area. The large area of the LV behind the PCDW VG is due to it having a larger frontal surface area than the surface of the PDW VG and the strength of the LV being introduced into the main flow. This causes the mixing of the fluid near the walls of the tubes with the fluid in the main flow to be better, meaning that the heat transfer rate increases [39]. In addition, the LV in the downstream area can compress the wake region so that the fluid flow velocity through the cylinder arrangement increases. This can increase the rate of heat transfer from the tube surface to the fluid flow in the wake region [37,39]. For this reason, the heat transfer enhancement brought about by using PCDWP VGs was better than that brought about by using PDWP VGs.

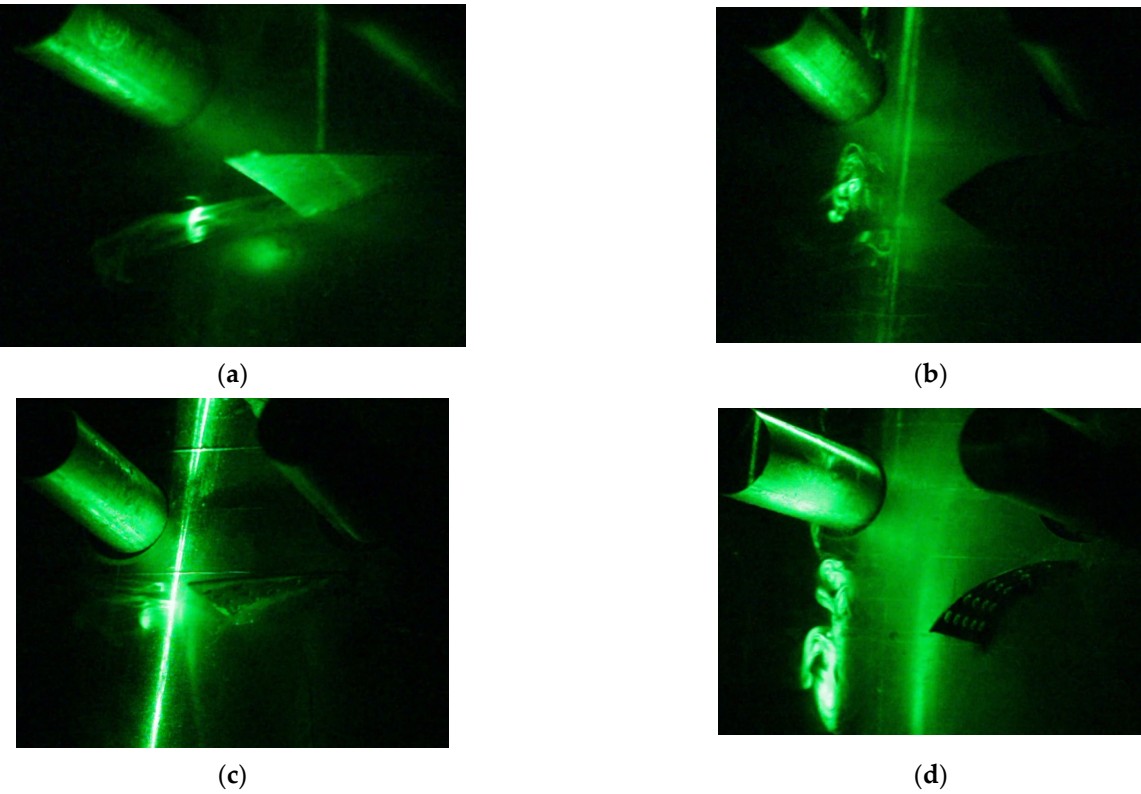

**Figure 8.** Visualization of the LV in the cross-section of the flow in the installation: (**a**) DWP, (**b**) CDWP, (**c**) PDWP, and (**d**) PCDWP VGs at the flow velocity of 1 m/s.

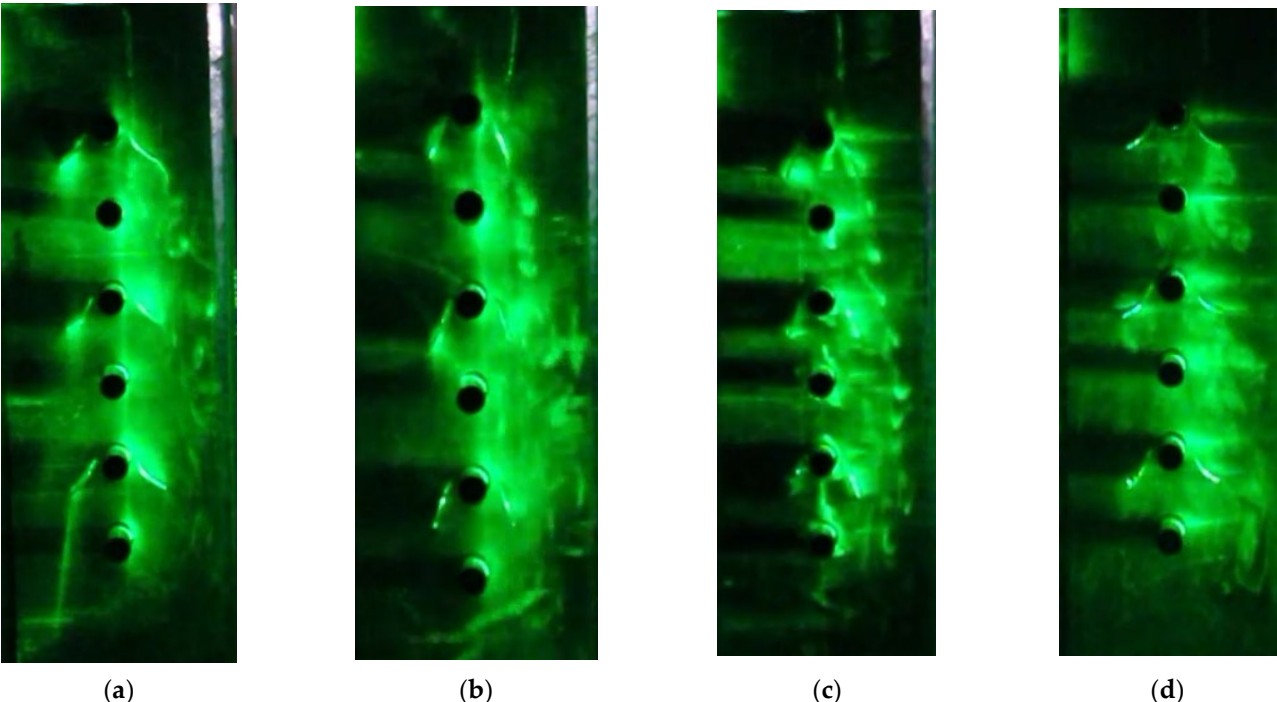

**Figure 9.** Flow structure in the unidirectional plane for installation: (**a**) CDWP, (**b**) DWP, (**c**) PDWP, and (**d**) PCDWP VG at the flow velocity of 1 m/s.

## 4. Uncertainty Analysis (Error)

### 4.1. Heat Transfer Uncertainty

In this study, the parameters of the mean, the standard deviation of the mean, and the overall error were used to determine the uncertainty [40,41]. This time, for the error calculation we used the test result data for the baseline case with a flow velocity of 0.4 m/s. Table 1 shows the tube surface temperature data from the test results for the baseline case.

**Table 1.** Tube surface temperature data from the baseline test at a flow velocity of 0.4 m/s.

| Data to | $T_{w1}$ (°C) | $T_{w2}$ (°C) | $T_{w3}$ (°C) | $T_{w4}$ (°C) | $T_{w5}$ (°C) | $T_{w6}$ (°C) |
|---------|---------------|---------------|---------------|---------------|---------------|---------------|
| 1 | 49.191 | 51.214 | 48.323 | 49.769 | 47.802 | 51.271 |
| 2 | 49.183 | 51.177 | 48.316 | 49.791 | 47.766 | 51.264 |
| 3 | 49.145 | 51.168 | 48.307 | 49.753 | 47.786 | 51.255 |
| 4 | 49.121 | 51.173 | 48.282 | 49.728 | 47.761 | 51.259 |
| 5 | 49.153 | 51.205 | 48.285 | 49.731 | 47.735 | 51.262 |
| 6 | 49.100 | 51.151 | 48.290 | 49.736 | 47.769 | 51.267 |
| 7 | 49.098 | 51.150 | 48.230 | 49.734 | 47.738 | 51.294 |
| 8 | 49.089 | 51.141 | 48.250 | 49.667 | 47.729 | 51.228 |

The data from Table 1 can be used to calculate the average tube surface temperature ($T_w$) using Equation (8) [40,41].

$$T_w = \frac{T_{w1} + T_{w2} + T_{w3} + T_{w4} + T_{w5} + T_{w6}}{6} \tag{8}$$

From Equation (8), we can obtain that the value of $T_w$ is 49.559 °C. Then, the average standard deviation can be calculated using Equation (9) [40,41].

$$s_{T_w} = \sqrt{\frac{\sum_{i=1}^{N} \left(T_{w_i} - T_w\right)^2}{N(N-1)}} \tag{9}$$

The value of $S_{T_w}$ is found to be 0.194 °C. Therefore, the value of $T_w$ can be written as 49.559 ± 0.194 °C. Meanwhile, $T_{in}$ and $T_{out}$ are found to be 28.887 °C and 32.954 °C, respectively. Then, the average standard deviation can be calculated using Equations (10) and (11) [40,41].

$$s_{T_{in}} = \sqrt{\frac{\sum_{i=1}^{N} \left(T_{in_i} - T_{in}\right)^2}{N(N-1)}} \tag{10}$$

$$s_{T_{out}} = \sqrt{\frac{\sum_{i=1}^{N} \left(T_{out_i} - T_{out}\right)^2}{N(N-1)}} \tag{11}$$

Therefore, $S_{T_{in}}$ and $S_{T_{out}}$ are found to be 0.012 °C and 0.045 °C, respectively. Furthermore, the values of $T_{in}$ and $T_{out}$ can be written as 28.887 ± 0.012 °C and 32.954 ± 0.045 °C, respectively. The standard deviation values of $T_{in}$ and $T_{out}$ are still higher than the accuracy value of the thermocouple used in the measurement. Additionally, the value of $Q$ at a flow rate of 0.4 m/s is found to be 19.477 W. Equation (12) can be used to calculate the error value of Q [40,41].

$$E_{a_{rss}Q} = \sqrt{\left(\Delta T_{in} \frac{\partial Q}{\partial T_{in}}\right)^2 + \left(\Delta T_{out} \frac{\partial Q}{\partial T_{out}}\right)^2} \tag{12}$$

where $\frac{\partial Q}{\partial T_{in}} = \frac{\dot{m} \cdot C_p \cdot (T_{in} - T_{out})}{T_{in}}$, while $\frac{\partial Q}{\partial T_{out}} = \frac{\dot{m} \cdot C_p \cdot (T_{in} - T_{out})}{T_{out}}$. With $\Delta T_{in} = 0.012$ °C and $\Delta T_{out} = 0.045$ °C, $E_{a_{rss}Q}$ is obtained at about 0.028 W. Therefore, the heat transfer rate was found to be $Q = 19.477 \pm 0.028$ W. This deviation of $Q$ is higher than the accuracy

of the Wattmeter used in the measurement. Furthermore, the *LMTD* is determined using Equation (4) to have a value of about 18.564 °C. Using Equation (13), the error value of *LMTD* is found to be 0.077 °C [40,41].

$$E_{a_{rss}\,LMTD} = \sqrt{\left(\Delta T_{in}\frac{\partial LMTD}{\partial T_{in}}\right)^2 + \left(\Delta T_w\frac{\partial LMTD}{\partial T_w}\right)^2 + \left(\Delta T_{out}\frac{\partial LMTD}{\partial T_{out}}\right)^2} \tag{13}$$

Then, the *LMTD* value can be written as 18.564 ± 0.077 °C. Furthermore, *Nu* is found to be 155.308 at a flow velocity of 0.4 m/s using Equation (2). The error for *Nu* is determined using Equation (14) [40,41]:

$$E_{a_{rss}\,Nu} = \sqrt{\left(\Delta LMTD\frac{\partial Nu}{\partial LMTD}\right)^2 + \left(\Delta Q\frac{\partial Nu}{\partial Q}\right)^2} \tag{14}$$

From Equation (14), the *Nu* error is found to be around 0.684. Therefore, *Nu* for the baseline case with a flow velocity of 0.4 m/s is 155.308 ± 0.684. Then, the value of the convection heat transfer coefficient is calculated using Equation (15) and is found to be 44,857 W/m$^2$K [40,41].

$$E_{a_{rss}\,h} = \sqrt{\left(\Delta Nu\frac{\partial h}{\partial Nu}\right)^2} \tag{15}$$

The error of the convection heat transfer coefficient is found to be 0.197 W/m$^2$K. Furthermore, the value of the convection heat transfer coefficient can be written as $h = 44.857 \pm 0.197$ W/m$^2$K. Using Equation (16), the percentage error of the convection heat transfer coefficient is found to be 0.44% [40,41].

$$\% \, Error_h = \left(\frac{E_{a_{rss}\,h}}{h}\right)100\% \tag{16}$$

The errors for all cases are determined in the same way. Therefore, the error percentage of the Nusselt number in the use of DWP, CDWP, PDW, and PCDW can be inferred from Table 2. From Table 2, it is found that the perforated VG installation error is higher than that without holes when using one and two VG pairs. The opposite is shown in the use of three VG pairs, where the error from the perforated VG installation is lower than that for the installation without holes.

**Table 2.** Overall error Nusselt number for the installation of DWP, CDWP, PDWP, and PCDWP VGs.

| Number of Pair | Overall Error | | | |
|:---:|:---:|:---:|:---:|:---:|
| | **DWP** | **CDWP** | **PDWP** | **PCDWP** |
| 1 | 1.05% | 0.60% | 1.69% | 0.90% |
| 2 | 0.67% | 0.75% | 0.77% | 0.74% |
| 3 | 1.12% | 1.15% | 0.85% | 0.91% |

*4.2. Uncertainty of Pressure Drop*

The data deviation of the pressure drop was determined based on the baseline case. The pressure drop data for the baseline case at a flow velocity of 2 m/s can be inferred from Table 3. Furthermore, the average pressure drop was found to be 0.013 in H$_2$O, which was calculated using Equation (17) [40,41].

$$\Delta P = \frac{\Delta P_1 + \Delta P_2 + \Delta P_3 + \cdots + \Delta P_{30}}{30} \tag{17}$$

**Table 3.** Pressure drop data from the baseline case at a velocity of 2 m/s.

| Pressure Drop (in $H_2O$) | | | |
|---|---|---|---|
| **Data to** | $\Delta P$ | **Data to** | $\Delta P$ |
| 1 | 0.013 | 16 | 0.012 |
| 2 | 0.013 | 17 | 0.013 |
| 3 | 0.013 | 18 | 0.012 |
| 4 | 0.013 | 19 | 0.012 |
| 5 | 0.012 | 20 | 0.013 |
| 6 | 0.013 | 21 | 0.013 |
| 7 | 0.013 | 22 | 0.012 |
| 8 | 0.012 | 23 | 0.013 |
| 9 | 0.013 | 24 | 0.012 |
| 10 | 0.013 | 25 | 0.013 |
| 11 | 0.013 | 26 | 0.013 |
| 12 | 0.013 | 27 | 0.013 |
| 13 | 0.012 | 28 | 0.013 |
| 14 | 0.012 | 29 | 0.012 |
| 15 | 0.013 | 30 | 0.012 |

Then, the mean standard deviation is calculated using Equation (18), obtaining $2.1 \times 10^{-5}$ in $H_2O$ [40,41].

$$s_{\Delta P} = \sqrt{\frac{\sum_{i=1}^{N} (\Delta P_i - \Delta P)^2}{N(N-1)}} \tag{18}$$

Therefore, the pressure drop value for the baseline case at a flow velocity of 2 m/s is $0.013 \pm 4.9 \times 10^{-5}$ in $H_2O$. Then, the error of the pressure drop is calculated by Equation (19) and is found to be 0.47% [40,41].

$$\% \; Error_{\Delta P} = \left( \frac{s_{\Delta P}}{\Delta P} \right) 100\% \tag{19}$$

The same method was used to calculate the error for all cases, as shown in Table 4. The error in the pressure drop measurement was higher in the case of the perforated VGs than in the case of there being no holes for the total number of VG pairs.

**Table 4.** Overall error of pressure drop for the installation of DWP, CDWP, PDWP, and PCDWP VGs.

| Number of Pair | Overall Error | | | |
|---|---|---|---|---|
| | **DWP** | **CDWP** | **PDWP** | **PCDWP** |
| 1 | 3.52% | 2.90% | 3.74% | 3.56% |
| 2 | 2.41% | 1.50% | 3.32% | 2.61% |
| 3 | 0.87% | 1.10% | 2.11% | 1.91% |

## 5. Conclusions

In this study, the evaluation of thermo-hydraulic performance was carried out in order to obtain the best performance. From the results of this experiment, we can draw the following conclusions:

1. The holes in the VG slightly reduce the heat transfer rate, which can be indicated by a decrease in the Nusselt number ratio of up to 9 to 10%.
2. The holes in the VG, in addition to slightly reducing the heat transfer rate, are also able to reduce the flow resistance, which can be indicated by a decrease in the friction factor ratio of up to 11 to 14.5%.
3. The thermal–hydraulic performance expressed as TEF with holes in the VG was slightly decreased by about 2.1% from that of with holes.

4. From an economic point of view, the use of perforated VGs can reduce the economy of a heat exchange system. This can be indicated by a threefold increase in CBR at the highest flow velocity for the three pairs of VGs.
5. The flow visualization showed the formation of a longitudinal vortex in the cross-section plane downstream of the VG.
6. The results of the study also showed that the data uncertainty obtained from calculating the heat rate and pressure drop was very low, with an average error of below 2% and 4% for *Nu* and $\Delta P$, respectively.

In the future, I will evaluate the installation of PCDW in in-line and staggered tube configurations and visualize them.

**Author Contributions:** Conceptualization, S.; methodology, S.; formal analysis, T.W.; investigation, T.W.; data curation, B.Y.; writing—original draft preparation, S.; writing—review and editing, S.; Reviewing the study results, N.S. All authors have read and agreed to the published version of the manuscript.

**Funding:** This study was funded by Faculty of Engineering of Diponegoro University (Dana Hibah RKAT), Indonesia, with contract number:/UN7.5.3.2/PL/2021.

**Data Availability Statement:** Not applicable.

**Acknowledgments:** This work was supported by the Faculty of Engineering of Diponegoro University (Dana Hibah RKAT), Indonesia, with contract number:/UN7.5.3.2/PL/2021. The authors are grateful to all research members, especially Lab. of Thermofluid of Mechanical Engineering of Diponegoro University Indonesia.

**Conflicts of Interest:** The authors declare no conflict of interest.

## Abbreviations

| | |
|---|---|
| $A$ | tube surface area (m$^2$) |
| $W$ | test section width (m) |
| $H$ | test section height (m) |
| $D$ | tube diameter (m) |
| $k$ | fluid thermal conductivity (W/mK) |
| $D_h$ | hydraulic diameter (m) |
| $\rho$ | density of air (kg/m$^3$) |
| $C_p$ | fluid specific heat (J/kgK) |
| $L$ | length of the test specimen plate (m) |
| $Q$ | heat transfer from the tube surface to the fluid flow (W) |
| $h$ | convection heat transfer coefficient (W/m$^2$ K) |
| $Nu$ | Nusselt number ($-$) |
| $Nu_0$ | Nusselt number for baseline ($-$) |
| $LMTD$ | log-mean temperature difference (K) |
| $T_{in}$ | inlet temperature of the fluid (K) |
| $T_{out}$ | outlet temperature of the fluid (K) |
| $T_w$ | wall temperature of the tube (K) |
| $V$ | air flow velocity (m/s) |
| $\dot{m}$ | fluid mass rate (kg/s) |
| $\Delta P$ | pressure drop along the flow (Pa) |
| $f$ | friction factor ($-$) |
| $f_0$ | friction factor for baseline ($-$) |
| $N$ | number of data ($-$) |
| $s_T$ | average standard deviation ($-$) |
| $E_a$ | absolute error ($-$) |
| TEF | thermal enhancement factor |
| CBR | cost–benefit ratio |
| VG | vortex generator |
| LV | longitudinal vortex |

| TV | transverse vortex |
|---|---|
| RW | rectangular winglet |
| DW | delta winglet |
| CDW | concave delta winglet |
| PDW | perforated delta winglet |
| PCDW | perforated concave delta winglet |
| RWP | rectangular winglet pair |
| DWP | delta winglet pair |
| CDWP | concave delta winglet pair |
| PDWP | perforated delta winglet pair |
| PCDWP | perforated concave delta winglet pair |

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
