# Peer review of "Evaluation of Vortex Generators in the Heat Transfer Improvement of Airflow through an In-Line Heated Tube Arrangement"

_fluids, doi:10.3390/fluids6100344_

Round 1

Reviewer 1 Report

The authors provide a study on Evaluation of Vortex Generators in Heat Transfer Improvement of Airflow through In-Line Heated Tubes Arrangement. The manuscript is well written, and carries information to the readers. I have, however, some fundamental remarks that if satisfactorily addressed could improve the final impact of the work.

1-The article does not completely contain a Nomenclature table. Please include all symbols in a table dedicated to them, which will be at the beginning of the article after the abstract and keywords. The description of each symbol (e.g., after each equation) may be avoided if a Nomenclature is provided, otherwise, all symbols should be clearly defined at the first instance of appearance in the manuscript.                                                     2-Better stress the connection between methods, linked to fundamentals, and the applied end side.                                                                                3-The novelty of this study is not clear, and needs to be highlighted explicitly to ensure it is clear to the reader what is new in the research and advancing the state of the art. Simultaneously, it needs to be made clear that the research does not cover only previously known information, and the knowledge gap needs to be clearly addressed.                                              4-The literature review is needed to be improved to meet the standard of the journal. The literature review does not appear adequate and it does not include sufficient important journal. It is highly recommended to read and use the following references in this section. doi.org/10.1016/j.icheatmasstransfer.2016.04.029. doi.org/10.1016/j.icheatmasstransfer.2013.04.011. doi.org/10.1115/1.4022994                                                                              5- Please provide appropriate references for the mentioned equations. Define all the parameters used in equations.                                                  6-Experimental analysis should be described in detail.                                      7-In experimental section, add the accuracy of measuring tools. What is the accuracy/error associated with experimental instruments and experimental data results? How many times the experiments conducted? More discussion is needed.                                                                                                        8-The uncertainty of the experimental data should be estimated in detail.    9-Please determine your future studies in some sentences in conclusion or discussion, to show readers how you want to proceed this work in the future.

Author Response

Thank you very much for your review of my paper. I have revised all things that you have suggested.

Reviewer 2 Report

The subject of the paper is quite interesting, but it is difficult to read the paper. Authors use notation, which is not clear. It causes that is very difficult to read the paper and appreciate its value. The main correction of the paper should be done to reedit text. 

The detailed remarks are included in added file.

Author Response

(The authors gave the same response as above.)

Round 2

Reviewer 1 Report

The revised version of the manuscript is good and it looks ready for publication.

Author Response

Thank you very much for your valuable comments

Reviewer 2 Report

The comments are included in the file.

Author Response

I have revised and answer all things that ask by the reviewer. The paper revision has been submitted into the journal.
